# Optimisation of cell and ex vivo culture conditions to study vascular calcification

**Nathalie Gayrard**[ID][1,2]*, **Karen Muyor**[1,2], **Cécile Notarnicola**[3], **Flore Duranton**[1,2], **Bernard Jover**[ID][3], **Àngel Argilés**[1,2,4]

**1** RD – Néphrologie 2, rue de Mûriers, Montpellier, France, **2** RD – Néphrologie, EA - 7288 BC2M BatD-RDC UFR - Pharmacie, Montpellier, France, **3** PhyMedExp (Physiologie et Médecine Expérimentale Cœur Muscles), INSERM-CNRS-Université Montpellier, IURC, Montpellier, France, **4** Néphrologie Dialyse Saint Guilhem (NDSG), CS 40 339, Sete, France

* Nathalie.gayrard@umontpellier.fr

## Abstract

Medial vascular calcification (MVC) is a highly prevalent disease associated with a high risk of severe, potentially lethal, complications. While animal studies may not systematically be circumvented, in vitro systems have been proven useful to study disease physiopathology. In the context of MVC, the absence of a clinically relevant standardized in vitro method prevents the appropriate comparison and overall interpretation of results originating from different experiments. The aim of our study is to establish in vitro models mimicking in vivo vascular calcification and to select the best methods to unravel the mechanisms involved in MVC. Human aortic smooth muscle cells and rat aortic rings were cultured in different conditions. The influence of fetal calf serum (FCS), alkaline phosphatase, phosphate and calcium concentrations in the medium were evaluated. We identified culture conditions, including the herein reported Aorta Calcifying Medium (ACM), which allowed a reproducible and specific medial calcification of aortic explants. Studying cells and aortic explants cultured, the involvement of bone morphogenetic protein 2 (BMP2) pathway, fibrosis and apoptosis processes in in vitro MVC were demonstrated. Expression of osteoblastic markers was also observed suggesting the occurrence of transdifferentiation of smooth muscle cells to osteoblasts in our models. The use of these models will help researchers in the field of vascular calcification to achieve reproducible results and allow result comparison in a more consistent way.

## Introduction

Medial vascular calcification (MVC), also known as Mönckeberg's sclerosis, occurs with aging in the general population and may also concern young adults with very prevalent diseases such as diabetes[1] and chronic kidney disease (CKD)[2,3]. To better understand MVC, the VDN rat (hypervitaminosis D and nicotine gavage) was set and showed aortic wall calcification with increased pulse pressure closely mimicking MVC in the human situation[4]. Other vascular calcification models associated with CKD as the 5/6 nephrectomy rat[5] or with diabetes as streptozotocin rat model[6] are also used. The use of animal models requires killing of a significant number of individuals. Ethical practices recommend reducing the use of in vivo models and replacing them as much as possible with alternative models according to the 3Rs rules

**Data Availability Statement:** The datasets are in supporting information file S2 Table data.

**Funding:** This project has received funding from French Languedoc Roussillon region's ARPE program under number 2014-005987 (BJ). The

funders had no role in study design, data collection and analysis, decision to publish, or preparation of the manuscript.

**Competing interests:** The authors have declared that no competing interests exist.

(replacement, reduction and refinement) elaborated in 1959 by W.M.S. Russell and R.L. Burch [7,8].

In keeping with these aims, vascular calcification is explored by in vitro techniques mainly involving two models, vascular smooth muscle cells (VSMC) and aortic rings from different species, commonly human, bovine, mouse and rat. Both cells and aortic rings are cultured with a variety of different calcifying media[9–12]. Most of these calcifying media include a high phosphate concentration. Beta glycerophosphate[9] or phosphate salts[10] are used as phosphate donors. High phosphate concentration parallels the levels observed in CKD patients, known to be prone to MVC development. Indeed, high phosphate level would favor matrix mineralization by VSMCs[13]. Vascular calcification as pointed out by Villa-bellosta is a very complex process where a passive and active deposition could be coexist[14]. Uremic vascular calcification in patients with end stage renal disease is partially mediated by elevated calcium x phosphorus product and is a regulated process who induces phenotypic modification of vascular cells into osteoblast-like cells[15]. The coexistence of passive and active processes in MCVpassive and active processes cannot be excluded. However, depending on medium composition and culture conditions, a high concentration of phosphate may also lead to massive phosphocalcic deposition in culture, which is not a mechanism observed in MVC. Osteoblastic differentiation of VSMCs is widely accepted as a major event in vascular calcification [16]. In this process, cells loose smooth muscle cell markers such as SM22α and smooth muscle α-actin, and express osteoblastic genes such as osteopontin or osteocalcin[17].

The methods and systems used to asses MVC involve a variety of pathogenetic mechanisms, some of which, particularly in the proposed "in vitro" systems could reflect only passive deposition and may be unrelated to those observed in human disease conditions. The aim of our study was to establish the most appropriate in vitro models mimicking as much as possible the vascular calcification observed in vivo and to select the best methods to unravel the mechanisms involved in MVC. In particular, we have tested different concentrations of serum, phosphate and calcium and assessed the influence of factors known to enhance calcification such as alkaline phosphatase.

## Methods

### Animals

The present animal experiments complied with the European and French laws (permit numbers D34-172-25 and 34179) and conform to the Guide for the Care and Use of Laboratory Animals published by the NIH (National Academies Press US, 8th edition, 2011). This study was approved by the local ethics committee "Comité d'éthique pour l'expérimentation animale Languedoc Roussillon N˚36". Six week-old Wistar rats (Charles River Laboratories) were housed in climate controlled conditions with a 12h light/dark cycle in a temperature-controlled room ($22\pm1$˚C). Rats were fed normal rat chow (A04, UAR) and tap water ad libitum. After one week of acclimatation, rats were anesthetized with isoflurane (2% in $O^2$) and then killed by decapitation. Aortas were removed, gently cleaned up, and 2–3 mm rings were cut for culture.

### Cell culture

Human aortic smooth muscle cells (HASMCs) were grown in smooth muscle cell medium (Science Cell). After the cells were grown to confluence, the smooth muscle cell medium was replaced with serum-free Dulbecco's modified Eagle's medium (Sigma-Aldrich) for 24h. Then cells were cultured 2, 3, 7 or 14 days and the basal medium is the Dulbecco's modified Eagle's medium containing 4.5 g.L$^{-1}$ glucose, 10 mM sodium pyruvate and 50 mg.mL$^{-1}$ ascorbic acid

(Sigma). Different concentrations of fetal calf serum (FCS, Biowest), calcium (Sigma-Aldrich), phosphate (Sigma-Aldrich) and alkaline phosphatase (Promega) were used to induce calcification.

### Ex vivo culture

The thoracic aortas were harvested from the descending part of the aortic cross to the diaphragm. The adjacent connective tissue was gently removed and the aorta was submitted to three successive PBS washings. Aorta was cut in rings of around 2mm thickness and aortic rings were cultured in 24 well plates with culture media and conditions as described above for cell culture.

### Measurement of total calcium content in tissue

Aortic rings were dehydrated at 110 ˚C. Cells or dried aortic rings were decalcified with 1mL and 100μL of 0.6 M of HCl, respectively for 24 h. Calcium content was measured in the supernatants using the O-cresolphthalein complexone method according to the instructions provided by the supplier (Biovision). HASMCs were solubilized in 1mL of 0.1M NaOH/0.1% SDS for the protein content determination by the bicinchoninic acid (BCA) protein assay kit (Pierce). Results were normalised to dry weight for aortic rings or protein content for cells.

### Histological staining

Segments of aorta were fixed in formalin and embedded in paraffin. Five micrometer-thick sections were cut and mounted on glass slides. Some slides were deparaffinized and stained with silver nitrate (Sigma-Aldrich) and counterstained with nuclear Fast Red (Sigma-Aldrich) for evaluation of calcium precipitation by von Kossa staining. Other slides were deparaffinized too and stained with Sirius red (Sigma-Aldrich) for evaluation of collagen localization. Positive signal for collagen was seen in intense red. The sections were mounted in Entelan (Merck) mounting medium and examined under a light microscope (Nikon Eclipse TE300). For each sample, quantification was performed with Image J software on pictures taken from 3 aorta sections at 40-fold magnification or 10 fields of HASMCs at 200-fold magnification. The staining area was measured on aortic medial layer and HASMC fields.

### Immunohistochemical staining

Immunohistochemical analyses of aorta were performed using paraffin sections. The primary antibodies, anti-osteopontin (1:100, Immuno-Biological Laboratories), anti-osteocalcin (1:100, Abcam) and anti phospho-Smad1 (1:200, Cell Signaling) were incubated overnight at 4˚C. Revelation was performed with Universal vectastain ABC kit and ImmPACT AEC (Vector Laboratories) following the instructions of suppliers. Then, the sections were mounted in an aqueous mounting medium, VectaMount ™ AQ (Vector Laboratories) and examined under a light microscope (Nikon Eclipse TE300). Quantification was carried out as for histological staining.

### Apoptosis assay

Apoptotic nuclei in aortic wall were identified by TUNEL staining according to the manufacturer's instructions (DeadEnd™ Colorimetric TUNEL System, Promega, USA). Briefly, sections were deparaffinized, fixed in 10% buffered formalin (Sigma-Aldrich) and immersed in 20 μg. $mL^{-1}$ proteinase K. Sections were washed in PBS, re-fixed in 10% buffered formalin. After equilibrating, sections were incubated at 37˚C with biotinylated nucleotide and the recombinant terminal deoxynucleotidyl transferase (rTdT) enzyme. After washing, sections were

blocked in 0.3% $H_2O_2$ (Sigma-Aldrich). One section was processed in parallel without rTdT enzyme as negative control. Another section was treated with DNase 1 to cause DNA fragmentation and processed in parallel as positive control. Then, the sections were mounted in Entelan mounting media and examined under a light microscope (Nikon Eclipse TE300). Quantification was carried out as for histological staining.

### Gene expression

HASMCs were trypsinized after 2, 3 or 7 days of culture and total RNA was extracted using RNAeasy mini kit (Qiagen). Reverse transcription was performed using 1 µg total RNA, anchored oligo-dT and Verso Reverse transcriptase (Thermo Scientific) according to the supplier protocol. For real-time PCR, gene expression levels were measured using SYBR Green I dye chemistry on LightCycler 480 system (Roche Applied Sciences). PCR primers were designed using the LightCycler Probe Design software 2.0 and table of primers is in supplementary data (S1_Table). Expression levels were determined with the LightCycler analysis software (version 3.5) relative to standard curves. Data were represented as the mean level of gene expression relative to the expression of the reference gene GAPDH.

### Statistical analysis

All data are expressed as mean ± SEM and analyzed by Analysis of variance (ANOVA) or mixed models. Data transformation was used to approach homoscedasticity. Random effects were included to control for the clustering of cultured aortas obtained from the same animal. Differences between groups were detected with post-hoc Tukey-Kramer's test. P-values less than 0.05 were considered statistically significant.

## Results

### High FCS, phosphate, and calcium with phosphate promote calcification of HASMC

To induce calcification we used culture media adapted from those previously described[10,18]: basal medium, 3.8 mM $NaH_2PO_4$/$Na_2HPO_4$ as phosphate donor and 0 or 15% FCS. We used wells containing the same medium without cells as internal controls. The highest calcium deposition was observed in HASMCs cultured with high phosphate (3.8 mM) and 0% FCS (Fig 1). Calcium deposits decreased in HASMCs cultured with high phosphate (3.8 mM) supplemented with 15% FCS. The controls of the different mediums alone, in the absence of cells, also showed the presence of significant amounts of calcium deposits in high phosphate (3.8 mM) without FCS, whilst no calcium deposits were observed in high phosphate conditions with 15% FCS supplementation. These results demonstrate that at the concentrations of phosphate and calcium used, calcium deposits may be formed with no need for cellular participation via cell-unrelated mechanisms (probably precipitation) in the absence of serum and that addition of serum in the medium prevents these cell-unrelated deposits to appear (probably by inhibiting precipitation).

We then tested the influence of different phosphate donors in inducing calcification. Three of them were tested: one acidic ($NaH_2PO_4$), one with neutral pH ($NaH_2PO_4$ / $Na_2HPO_4$) and one with basic pH ($Na_2HPO_4$). Acidic, neutral and basic pH showed successively increasing calcium deposits on cells in terms of calcified surface areas as observed by von Kossa staining (Fig 2A, P < 0.01). The total calcium content in the wells was significantly increased in the wells with cells cultured with the basic P donor only (P < 0.001 vs acidic and neutral conditions). In the absence of cells, no calcium deposition was observed further suggesting that

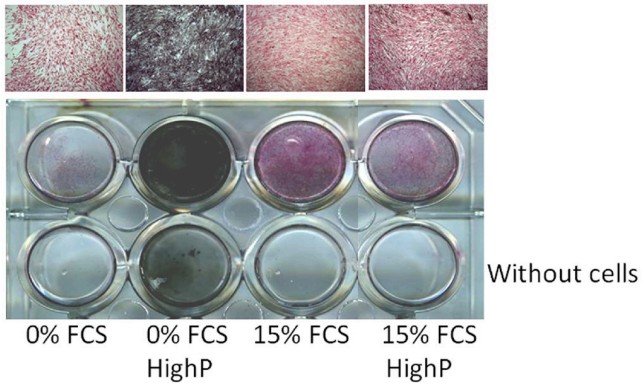

**Fig 1. Influence of Fetal Calf Serum (FCS) and phosphate concentrations on calcium deposition in HASMC.** Von Kossa staining after 14 days of culture with or without cells. No calcium staining was observed in low phosphate containing media. Calcium staining was observed in high P media, in the absence of FCS both, in the presence as well as in the absence of cells depicting a passive mechanism (no need for cells to obtain phosphocalcic deposition). In high phosphate concentration (3.8 mM) and FCS supplemented wells, no calcium was observed in the absence of cells whilst a restricted positive calcium staining was observed in the presence of cells, suggesting the involvement of an active process (cell related) in the appearance of the calcium deposits, expected to be more representative of MCV disease. (Original magnification x200).

passive precipitation of calcium does not exist in these three conditions with 15% supplemented FCS but the calcium deposit is very low in the wells with cells (Fig 2A).

With the aim to increase the yielding in terms of calcium deposits, we checked the influence of calcium concentration in the medium (Fig 2B). Increasing concentrations of calcium with a fixed high phosphate concentration resulted in increasing amounts of calcium deposits in cell culture, with a marked difference threshold at 2.4 mM (P < 0.001). Based on these results, 2.4 mM Ca was chosen for further studies (Fig 2B).

The effects of high calcium and/or high phosphate in the culture medium, as well as phosphate donor type on calcium deposit formation in cells were also tested (Fig 2C). The two factors were associated with calcification level assessed both, by von Kossa staining and calcium content, and interactions were present (all P < 0.001). High concentration of both phosphate and calcium did result in calcium deposit in HASMC cells (* P < 0.01 vs high P or high Ca with each P donor), with increasing intensity from acidic to neutral or basic P donor (P < 0.01). However, calcium deposition was also observed in wells without cells suggesting also the presence of a passive calcium deposition (or precipitation). The phosphate donor with the greatest ratio of active/passive calcium deposition was the neutral phosphate buffer ($NaH_2PO_4$ / $Na_2HPO_4$), which had the weakest calcium staining in the absence of cells and a strong calcium staining of the cells (Fig 2C).

## Effect of alkaline phosphatase on calcification of HASMC

We also tested the influence of alkaline phosphatase on calcium deposition in HASMC cultures. We used a medium containing 3.75 units $mL^{-1}$ of alkaline phosphatase as described previously[10] (Fig 2D). The calcium content was influenced by FCS (P < 0.001) and ALK (P < 0.001), and the effect of FCS varied with the presence of cells (P < 0.001) and ALK (P < 0.01). In the absence of cells and FCS, calcium deposition was very high in serum free media (p<0.003) and was not modified by the addition of ALK (p = 0.9); suggesting that the passive calcium deposition (absence of cells) is not modulated by ALK in serum free conditions. In cells cultured with 15% FCS supplemented media alkaline phosphatase clearly

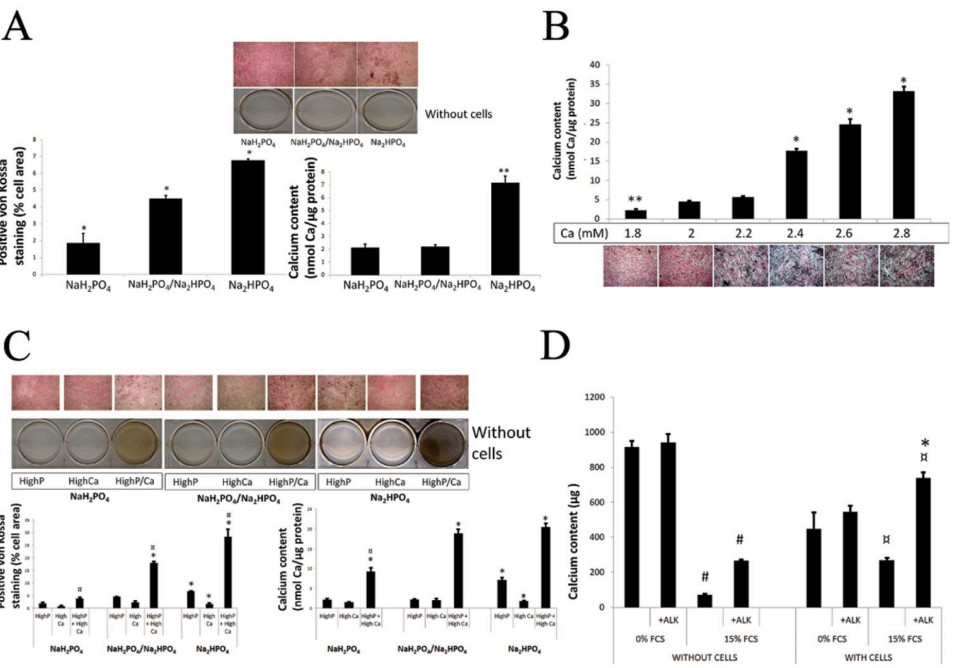

**Fig 2. Influence of cell culture medium composition on calcification of HASMC.** A) Influence of the phosphate donor in medium supplemented with 15%FCS. No calcium was observed in the absence of cells (wells of the lower panel). While significant amounts of Ca were observed in the cell cultures (upper panel). The higher calcium deposit on cells was observed with basic ($Na_2HPO_4$), compared to acidic ($NaH_2PO_4$) or neutral pH ($NaH_2PO_4/ Na_2HPO_4$) phosphate buffers (* $P<0.01$ versus others; ** $P<0.001$ versus others). B) Influence of calcium concentration in medium supplemented with 15%FCS and high phosphate (3.8 mM). Calcium deposition was proportional to the Ca content in the medium in the media with 3.8 mM phosphate (* $P<0.01$ versus others; ** $P<0.001$ versus others). C) Influence of phosphate donor, phosphate level and calcium level in medium supplemented with 15%FCS. High phosphate concentration (3.8 mM) alone induced a significant calcium deposition in cell culture, while high Ca (2.4 mM) alone did not. High phosphate associated with high Ca medium resulted in the highest calcium deposition; the positive Ca staining in the absence of cells (wells in the lower panels) suggests also the participation of a passive phosphocalcic deposition. Calcium deposition was again higher with basic phosphate donor ($Na_2HPO_4$) buffer compared to the other phosphate donors (* $P<0.01$ versus other P/Ca conditions with the same P donor; ¤ $P<0.001$ versus other P donors with the same P/Ca condition). D) Influence of Alkaline phosphatase (ALK) and FCS in high phosphate and high Ca medium. ALK increased calcium deposition in the presence of FCS, both in the presence of cells as well in the absence of cells (¤ $P<0.05$ for the effect of cells compared to the same FCS and ALK conditions; # $P<0.001$ for the effect of FCS compared to the same cell and ALK conditions; * $P<0.05$ for the effect of ALK compared to the same cell and FCS conditions).

increased calcium deposition ($P < 0.01$). The same was observed in the absence of cells in 15% supplemented media ($p<0.01$).

Taken together, these results suggest that FCS is required by alkaline phosphatase to exert its enhancing activity on calcium deposition and that this enhancing activity may be independent of the presence of cells (Fig 2D).

## Osteoblastic transdifferentiation of HASMC cultured in cell calcifying medium

Next we explored the impact of osteogenic pathway activation in our cell model. According to the previous results, osteoblastic lineage genes expression was assessed in HASMC cultured in calcifying medium (CM) characterized by high phosphate (3.8 mM) and calcium (2.4 mM) concentration, 15% FCS and neutral phosphate donor ($NaH_2PO_4 / Na_2HPO_4$). The expression

of BMP2 and BMPR1A assessed by qPCR was increased after 48 hours of culture in calcifying medium compared to control medium (Fig 3A, both P < 0.01). BMP2 signaling pathway was further explored by Western blot (Fig 3B). At days 3 and 7 of culture, activation of smad1 by phosphorylation was demonstrated by measuring the expression of phosphosmad1. In calcifying medium, phosphosmad1 expression was enhanced compared to control medium (Fig 3B, both P<0.01). Expression of osteoblastic genes was also assessed at day 7 of culture by qPCR (Fig 3C). Compared to the cells cultured in control medium, the cells cultured in calcifying medium had an increased the expression of osteoblastic genes RUNX2, RANKL, OPN, OPG and ALP.

## Influence of FCS, phosphate and calcium on aortic ring calcification "ex vivo" model

The other model frequently used to study vascular calcification involves ex vivo cultured aortic rings. As for HASMC culture, different culture media were tested: basal medium with 0% or 15% FCS, and basal (0.9 mM) or high phosphate (3.8 mM). Aortic rings, cultured in the presence of FCS displayed a positive von Kossa staining distributed alongside the medial layer of the arterial explant demonstrating mediacalcosis (Fig 4). No calcium deposits were observed in the aortic rings in the absence of FCS or in the FCS-supplemented culture medium at basal concentration of phosphate. Therefore, both FCS and high phosphate concentrations are required to induce calcification of the medial layer of the aortic explants.

Then, we assessed the influence of phosphate donors on medial calcification of aortic rings cultured in 15% FCS-supplemented medium (Fig 5A). Total calcium content was high and similar with the three tested phosphate donors (acidic ($NaH_2PO_4$), neutral ($NaH_2PO_4$ / $Na_2HPO_4$) and basic ($Na_2HPO_4$), P = 0.6). In contrast, von Kossa staining suggested differences in the positively stained surface (P = 0.08). Aortic rings cultured with $NaH_2PO_4$ / $Na_2HPO_4$ as phosphate donor displayed the highest surface area with labelled calcium in the medial layer (Fig 5A). The discrepancy between results obtained by calcium dosage and von Kossa staining relies in that these techniques measure different aspects: the von Kossa staining assesses the calcified surface of the medial layer of aortic rings, whilst the calcium assay evaluates the calcium content of the entire aortic ring.

We then assessed the individual and combined influence of phosphate and calcium level on aortic ring calcification. There was a difference in calcium content and positive staining with different phosphate and calcium levels in culture medium (Fig 5B, both P < 0.01). In basal phosphate concentration media (0.9 mM), increased concentrations of calcium (up to 2.4 mM) did not result in significant aortic ring calcification (Fig 5B). High phosphate concentration (3.8 mM) alone resulted in significant calcium content of aortic rings and large surface of the medial layer positively stained (P < 0.01); addition of 2.4 mM calcium was associated with higher calcium content in aortic rings (P < 0.001). However, stained calcium was mainly around the aortic ring and not in the medial layer (Fig 5B).

We also tested the effect of alkaline phosphatase on medial calcification of cultured aortic rings (Fig 5C). The addition of FCS increased calcium content and positively stained areas (P <0.001). Calcium content and von Kossa stained areas were not influenced by the addition of alkaline phosphatase in the absence of FCS (P = 0.9). In the presence of FCS, the addition of alkaline phosphatase influenced Calcium content (P = 0.008), and a similar tendency was observed in positive staining (P = 0.1). Contrasting with what was observed in cell cultures, alkaline phosphatase in FCS-supplemented media decreased instead of increased calcification. The optimal culture conditions to obtain medial calcification of aortic rings were 15% FCS-supplemented medium without alkaline phosphatase.

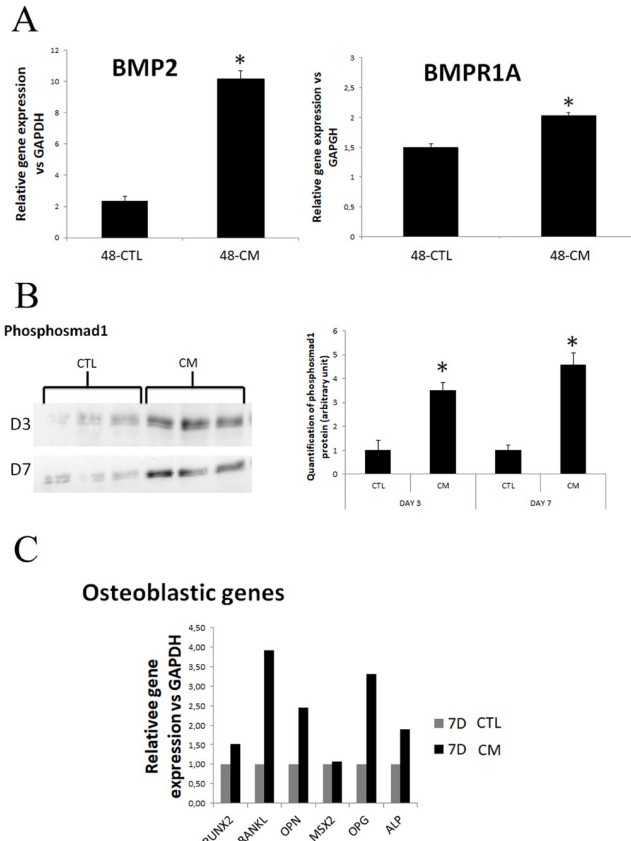

**Fig 3. Pro-osteogenic pathway exploration in HASMC cultivated in calcifying or control medium.** A) BMP2 and BMPR1A RNA expression in HASMCs by qPCR after 48 hours of culture. Both gene expressions were increased in cells cultured in calcifying medium (high phosphate, high calcium, high FCS) compared to the control medium. Gene expression was normalized to GAPDH expression. B) Exploring BMP2 signalling pathway by Western blot of phospho-Smad1 protein. The expression of phospho-Smad1 was increased in calcifying medium compared to control medium (* P<0.01 at day 3 and day 7). C) Relative quantification of gene expression by qRT-PCR analyses. The expression of runt-related transcription factor 2 (RUNX2), receptor activator of nuclear factor kappa-B ligand (RANKL), osteopontin (OPN), osteoprotegerin (OPG) and alkaline phosphatase (ALP) genes was increased in calcifying medium compared to control medium. (Gene expression was normalized to GAPDH expression; CTL: control medium; CM: calcifying medium).

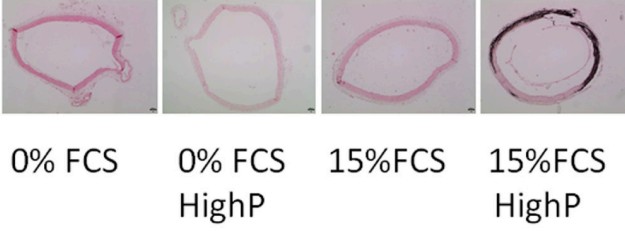

**Fig 4. Influence of Fetal Calf Serum (FCS) and phosphate concentrations on calcium deposition in aortic rings.** Von Kossa staining of aortic rings after 14 days of culture in the absence or presence of 15% FCS, both in basal medium or high phosphate (3.8 mM) medium. Note a clear von Kossa positive band following the distribution of the medial layer of the aortic ring only in the presence of FCS and high phosphate medium. (Original magnification x40).

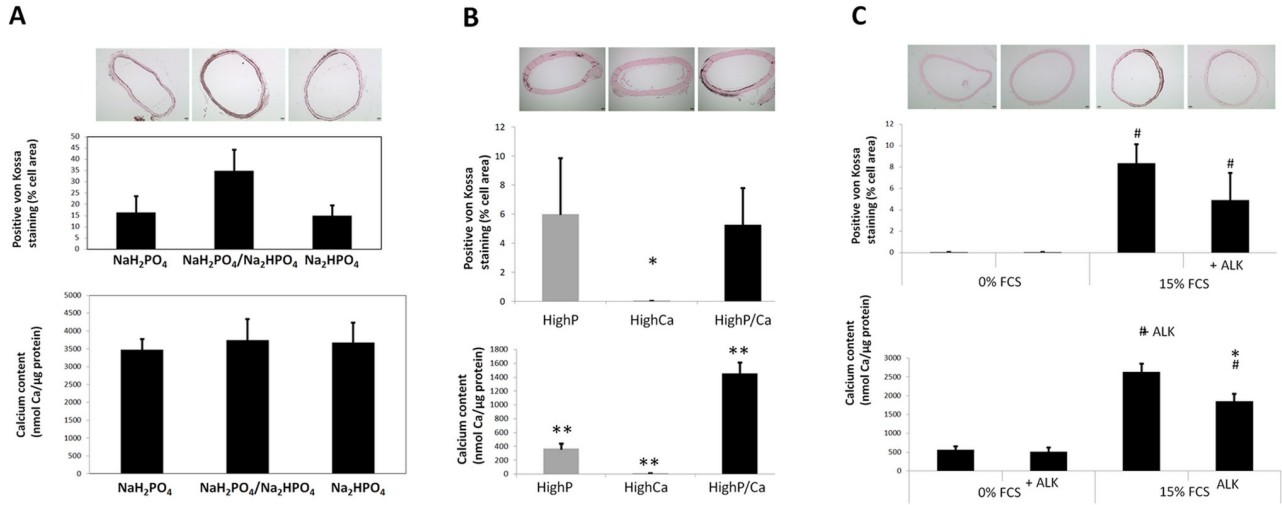

**Fig 5. Influence of medium composition on ex vivo calcification in aortic rings culture.** A) <u>Influence of the phosphate donor</u> (basic, neutral or acidic). Calcium incorporated into the tunica media of aortic rings assessed by von Kossa staining was higher with neutral phosphate donor buffer ($NaH_2PO_4$ / $Na_2HPO_4$), P = 0.08. B) <u>Effect of calcium and phosphate concentrations on calcification.</u> A marked increase in calcium content was observed in high phosphate/high calcium conditions, whilst no increase in von Kossa staining positive surface area was observed when compared with high phosphate, suggesting that a significant proportion of calcium was not incorporated into the tunica media (* P<0.01 versus others; ** P<0.001 versus others). C) <u>Effect of alkaline phosphatase on calcification.</u> In 15% FCS supplemented cultures, calcium deposition decreased in the presence of alkaline phosphatase (# P<0.001 for the effect of FCS compared to the same ALK condition; * P<0.01 for the effect of ALK compared to the same FCS condition). (original magnifications x40).

Following the results of all experiments presented above, we defined the best medium to obtain aortic media calcification as that composed of the basal medium described in methods supplemented with 15% FCS and 3.8 mM NaH2PO4 / Na2HPO4; it is referred to as aorta calcifying medium (ACM) in the subsequent experiments.

## Apoptosis, fibrosis and transdifferentiation in aortic rings cultured in aorta calcifying medium

Culture of aortic rings with ACM resulted in medial layer calcification associated with fibrosis (positively labelled Sirius red) and apoptosis (positive for TUNEL assays), Fig 6A. Finally, we assessed the presence of phosphorylated Smad-1 (as a marker of BMP2 pathway activation) and of osteopontin and osteocalcin (markers of osteoblastic transdifferentiation) by immuno-histochemistry. The three proteins were present in aortic rings cultured with ACM and not in the rings cultured with control medium (Fig 6B). This suggests that aortic rings undergoing calcification of the medial layer have an activation of the BMP2 pathway, an involvement of fibrosis and apoptosis processes and are submitted to cell transdifferentiation to osteoblastic phenotype.

## Discussion

To mimic vascular calcification, commonly observed in humans during the normal physiological life span (aging) and in pathological conditions such as chronic kidney disease, different models have been proposed. In vitro systems are preferred when the full animal is not strictly needed. The multiplicity of systems may render difficult comparison and extrapolation of the results of the different reports. The main aim of our study was to evaluate the different factors influencing the yielding and reproducibility of the in vitro calcification systems in order to

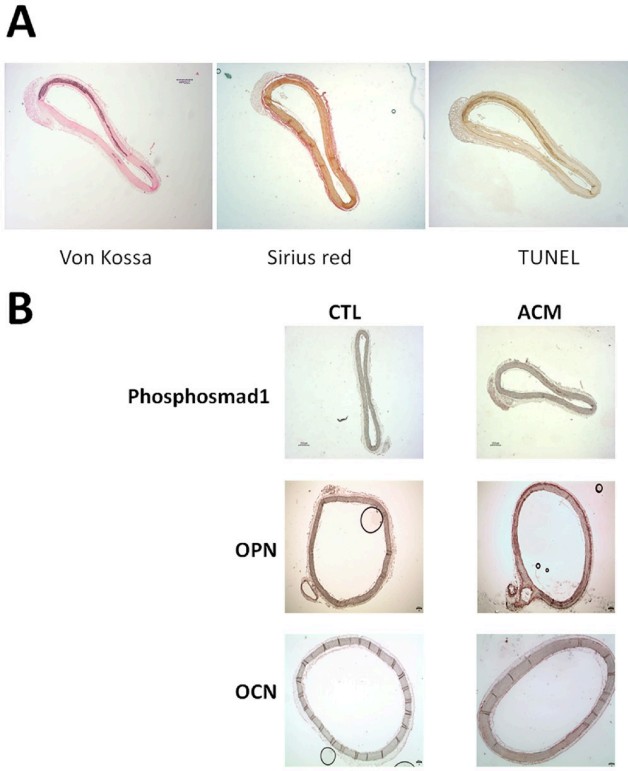

**Fig 6. Mechanisms participating in calcium deposition.** A) Calcium deposits, fibrosis and apoptosis: Aortic rings cultured in calcifying medium were positively stained with von Kossa, Sirius red and TUNEL staining, suggesting that fibrosis and apoptosis occur in calcium deposits induced ex-vivo in aortic rings in culture. B) Pro-osteogenic pathway exploration in ex-vivo aortic ring calcification in culture. Phospho-Smad1, Osteopontin and Osteocalcin expressions were observed in aortic rings cultured in calcifying medium (ACM), whilst were not observed in control medium (CTL) suggesting that calcification process involves BMP2 pathway and trans-differentiation to osteoblastic phenotype (original magnifications x40).

propose a method which might be used by several groups, allowing proper comparison of the data contributed by a wider community of researchers. Two widely used in vitro models of vascular calcification systems are presented and analysed: in vitro vascular smooth muscle cells and ex vivo aortic ring explants cultured in calcifying conditions. Our data demonstrate the influence of many factors on the calcium content and distribution in cells and tissues to be taken into account when performing calcification experiments, and even more importantly when interpreting the results of such experiments. We described the influence of the type of phosphate donor, the presence of FCS in the culture medium, the phosphate and calcium concentrations and the presence of alkaline phosphatase. In addition to individual effects, complex interactions between factors were present, which could vary between cell and aortic ring models. Use of internal controls, reproducibility and cautious interpretation of the results are main aspects to consider when using an in vitro system in the study of vascular calcification. Still, when adequately assessed, in vitro systems allow the characterization of phenomena participating in the calcification process, such as the involvement of the BMP2 pathway, the expression of osteoblastic genes and proteins, and changes in fibrosis and in apoptosis processes. The expression of BMP2 is upregulated in human arteries with medial calcification[19] and BMP2 is a promotor of calcification in high phosphate condition in vitro[20]. This is a major signaling pathway involved in vascular soft tissue mineralization[21–23].

In vitro techniques can be useful to study the implication of one protein or pathway, however they may also provide results that are unrelated or unimportant for vascular calcification as suggested by Hortells et al[24]. Our approach allowed identifying factors relevant in the processes observed in in vitro calcification systems, particularly by including internal controls in the absence of cells. By comparing results in presence and absence of cells, the relevance of a positive result may be estimated. This was the case when cultures were performed with a serum-free high phosphate medium and an increase in calcium content was also observed in the wells without cells in the culture system (Figs 1, 2C and 2D). Furthermore, by using two different quantification systems, i.e. total calcium content measurement by colorimetric assay as well as cellular or tissue distribution of calcium as assessed by von Kossa staining, the active participation of the cells and aortic tissue to the calcium deposit formation may be distinguished from simple calcium accumulation as assessed by calcium content. The pattern of accumulation in the medial layer of the aortic tissue closely mimics the distribution observed in human Mönckeberg's sclerosis and is strongly suggestive of active cellular participation in the vascular calcification process. The measurement of calcium content only by colorimetric methods may lead to overestimation of medial vascular calcification and inappropriate conclusions about processes involved in vascular calcification.

Most in vitro systems proposed to study vascular calcification utilize a high concentration of phosphate as phosphate is certainly an essential parameter.[10,12,17,25] Indeed, the product of serum calcium and phosphorous (Ca x $PO_4$), elevated in CKD patients, has been considered to be central in vascular calcification until recently.[16,26,27] Our study provides information on the independent participation of phosphate and calcium on vascular calcification. It is observed that with low phosphate, even high concentrations of calcium do not result in vascular calcification, confirming again the importance of phosphate in the calcification process. Thus, on this basis of these results and based on previous work we chose 3.8 mM (11,7 mg · $dL^{-1}$) phosphate in our systems to evaluate vascular calcification. With this fixed phosphate concentration, increasing calcium concentration in the culture medium resulted in a proportional increase in cellular calcium deposition (Fig 2B). Another aspect we assessed was the influence of the phosphate donor on vascular calcification yielding. We observed differences between the different phosphate donors tested, and identified the neutral phosphate donor as best suited for inducing vascular calcification in cell cultures in high calcium condition (2,4 mM or 9,6 mg · $dL^{-1}$) (Fig 2).

The use of FCS is of outmost importance. Since nonspecific calcium deposition was observed in wells in 0% FCS cell-free cultures (rather due to precipitation) and a very nice distribution of the calcium deposits along the medial layer was observed in aortic rings in the presence of FCS, it is clear that any active process leading to vascular calcification should be assessed in vitro in FCS-supplemented media. FCS also influences the effect of alkaline phosphatase on vascular calcification. In the cell culture system, alkaline phosphatase increased calcium deposits only in the presence of FCS (Fig 2D), and FCS was also required to observe the alkaline phosphatase effect in aortic ring culture. However, interestingly, the alkaline phosphatase effect was opposite to that observed in cell cultures: alkaline phosphatase decreased calcium deposition in aortic rings in the presence of FCS (Fig 5). Interestingly, our results are in contrast with those reported by Lomashvilli et al [28] that found an increase in calcification of aortic rings with partially removed adventitia and in the absence of FCS, further underlying the importance of FCS in the medium when analyzing the calcification in aortic explants, particularly to assess long-term cultures (over 14 days) and active processes. Based on these contradictory results between the two systems we tested, we decided not to use alkaline phosphatase in our subsequent studies on vascular calcification and recommend further studies to be performed accordingly.

In summary, the combination of results obtained with cell and aortic ring cultures enables to answer specific questions in vascular calcification studies provided that models are well controlled for all the parameters we have reported in the present study. In vitro systems allow working with human cells and obtaining enough biological material to perform all the procedures necessary to dissect the mechanisms participating in vascular calcification (Western blotting, qPCR and others). In addition, aortic ring cultures allow the identification of the spatial distribution of the calcium deposits and visualisation of the calcified area by von Kossa staining and the assessment of protein expression in the aorta using immunohistochemistry methods. However, caution is mandatory when trying to extrapolate the in vitro and ex vivo results to the in vivo situation. Although we share the aim of limiting as much as possible animal experimentation, the latter remains necessary to ultimately check whether what is observed in vitro does actually occur in the in vivo situation. This is a required step before transposing to human pathology.

## Supporting information

**S1 Table.**
(XLSX)

**S2 Table. Data.**
(XLSX)

**S3 Table. Culture conditions.**
(XLSX)

**S4 Table. Medium calcium concentration.**
(XLSX)

**S1 Fig. Raw images WB phosphosmad1 day 3 original images Fig 3B.**
(PDF)

**S2 Fig. Raw images WB phosphosmad1 day 7 original images Fig 3B.**
(PDF)

## Acknowledgments

We thank Montpellier's histology platform, the RHEM for its technical assistance.

## Author Contributions

**Conceptualization:** Nathalie Gayrard, Bernard Jover, Àngel Argilés.

**Formal analysis:** Nathalie Gayrard, Flore Duranton.

**Investigation:** Nathalie Gayrard, Karen Muyor, Cécile Notarnicola, Bernard Jover.

**Methodology:** Nathalie Gayrard, Cécile Notarnicola, Bernard Jover.

**Project administration:** Nathalie Gayrard.

**Supervision:** Nathalie Gayrard, Bernard Jover, Àngel Argilés.

**Validation:** Nathalie Gayrard, Bernard Jover.

**Visualization:** Nathalie Gayrard.

**Writing – original draft:** Nathalie Gayrard, Àngel Argilés.

**Writing – review & editing:** Nathalie Gayrard, Flore Duranton, Bernard Jover, Àngel Argilés.

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
