## [Decision Letter · Decision Letter 0]

30 Dec 2019

PONE-D-19-33486

Optimisation of cell and * ex vivo * culture conditions to study vascular calcification

PLOS ONE

Dear Dr gayrard,

Thank you for submitting your manuscript to PLOS ONE. After careful consideration, we feel that it has merit but does not fully meet PLOS ONE’s publication criteria as it currently stands. Therefore, we invite you to submit a revised version of the manuscript that addresses the points raised during the review process.

As you can see, all four reviewers were positive about the manuscript, but several minor changes were suggested, mostly requesting to provide more details in the methods, that could further strengthen the paper.

We would appreciate receiving your revised manuscript by Feb 13 2020 11:59PM. To enhance the reproducibility of your results, we recommend that if applicable you deposit your laboratory protocols in protocols.io, where a protocol can be assigned its own identifier (DOI) such that it can be cited independently in the future. For instructions see: http://journals.plos.org/plosone/s/submission-guidelines#loc-laboratory-protocols

We look forward to receiving your revised manuscript.

Kind regards,

Harald Mischak

Academic Editor

PLOS ONE

Journal Requirements:

Reviewers' comments:

Reviewer's Responses to Questions

**Comments to the Author**

1. Is the manuscript technically sound, and do the data support the conclusions?

Reviewer #1: Yes

Reviewer #2: Yes

Reviewer #3: Yes

Reviewer #4: Yes

2. Has the statistical analysis been performed appropriately and rigorously? 

Reviewer #1: Yes

Reviewer #2: Yes

Reviewer #3: Yes

Reviewer #4: Yes

3. Have the authors made all data underlying the findings in their manuscript fully available?

Reviewer #1: Yes

Reviewer #2: Yes

Reviewer #3: Yes

Reviewer #4: Yes

4. Is the manuscript presented in an intelligible fashion and written in standard English?

Reviewer #1: Yes

Reviewer #2: Yes

Reviewer #3: Yes

Reviewer #4: Yes

5. Review Comments to the Author

Reviewer #1: COMMENTS FOR AUTHORS

PlosONE

MANUSCRIPT # PONE-D-19-33486 REVIEWER # 1

Nathalie Gayrard

Optimization of cell and ex vivo culture 1 conditions to study vascular calcification

The authors of this manuscript establish in vitro models mimicking in vivo vascular calcification and to select an optimal method to unravel the mechanisms involved in medial vascular calcification. The authors identified culture conditions, which allowed a reproducible and specific medial calcification of aortic explants as well as calcification of human aortic smooth muscle cells. The authors concluded these approaches will help to achieve reproducible results and allow result comparison in a consistent way.

The data presented in this manuscript are interesting for terms of analyses of vascular calcification processes. However, the reviewer has some major remarks:

Major points:

1. The introduction section has to be revised. The authors should focus on the main topic of the manuscript and should avoid an very general description of the methods available for the investigation of vascular calcification processes (eg.g page 3 , line 56-59).

2. The authors should prepare a table demonstrating the condition analyze within the study.

3. NG, KM, FD and AA are obviously employees of a company (RD-Nephrologie). A statement has to be added whether these authors have to declare a conflict of interest based on the employment.

4. Figure 3.B: A bar graph demonstrating the quantification of Phosphosmad1 should be added.

5. Not significant data has to be removed from the revised version of the manuscript (page 12, line 258 - 260)

6. The authors should discuss the overall aim of the study in the discussion section in more detail.

7. Some typos should be removed in the revised version of the manuscript, e.g. page 9, line 169 (3.8mM instead of 3.8 mM).

Reviewer #2: This experiment allows overcoming of an ethical issue for using animal models in experiments of vascular calcification research as one of the most important outcome for experiencing major cardiovascular events in humans as a topic for research, although the authors well pointed out the extrapolation of the data should be taken carefully into account.

Reviewer #3: The aim of this study is to analyze in vitro and ex vivo models of vascular calcification and to select factors that may affect the process of calcification.

Th study is well performed and I cannot identify methodological defects

However there are points that should be clarified

-Fig1 shows that calcium deposits in HASMCs cultured with high phosphate

(3.8 mM) are decreased by supplementing with 15% FCS. It is important to determine if ionized calcium and the concentration of phosphate are modified by 15% FCS. It is know that FCS contains large proteins “calciproteins” that may substract Ca and P from the solution. At least the authors should evaluate if for a given pH, FCS modifies the Ca x P product.

-The total calcium content in the wells was significantly increased in the

wells with cells cultured with the basic P donor only (P < 0.001) vs acidic and neutral

conditions. It is likely that the high pH is increasing calcium deposit the opposed is expected to occur in acid pH. Again ionized Ca and pH should be measured minute after the addition of the different forms of phosphate.

-It seems that FCS is required by alkaline phosphatase to increase activity on calcium deposition and that this enhancing activity may be independent of the presence of cells (Fig 2D). It will be important to analyze if FCS contains pyrophosphate; hydrolysis of pyrophosphate by alkaline phosphatase may affect calcification

-The authors state the the optimal culture conditions to obtain medial calcification of aortic rings were 15% FCS supplemented medium without alkaline phosphatase. Thus data should be contrasted with results by KI 2008 73(9):1024 .

Reviewer #4: The authors address a key problem in calcification research, which is the lack of reproducibility of in vitro calcification set ups. I have only minor comments.

1. What was the cell culture media pH for different phosphate donors used to induce calcification?

2. Please indicate the number of times that experiments were repeated and provide DS for graphs without DS

3. Specifically, for the optimal ACM, please indicate whether every time the experiment was repeated, calcification was obtained or whether there are still some uncontrolled factors that may prevent calcification in certain experiments

4. Please in the discussion, add the concentrations of calcium and phosphate used in mg/dl, which are commonly used units in many countries, so as readers can easily contextualize to the clinical situation. Since calcium binds albumin and albumin levels in 15% FCS are lower than in 100% human serum, please discuss what would be the free calcium and total calcium equivalent of the concentrations used in experiments.

6. PLOS authors have the option to publish the peer review history of their article (what does this mean?). If published, this will include your full peer review and any attached files.

Reviewer #1: Yes: Joachim Jankowski

Reviewer #2: No

Reviewer #3: No

Reviewer #4: No

---

## [Author Response · Author response to Decision Letter 0]

14 Feb 2020

Dear Editor,

Please find enclosed the response to reviewers. 

We checked the style requirements

The complete data are available in supplemental data, S2_Table Data

Blot/gel image data are in Supporting Information, S5_raw_images WB phosphosmad1 day 3 original images Figure 3B.pdf and S6_raw_images WB phosphosmad1 day 7 original images Figure 3B.pdf

Ok

Reviewer #1: COMMENTS FOR AUTHORS

PlosONE

MANUSCRIPT # PONE-D-19-33486 REVIEWER # 1

Nathalie Gayrard

Optimization of cell and ex vivo culture 1 conditions to study vascular calcification

The authors of this manuscript establish in vitro models mimicking in vivo vascular calcification and to select an optimal method to unravel the mechanisms involved in medial vascular calcification. The authors identified culture conditions, which allowed a reproducible and specific medial calcification of aortic explants as well as calcification of human aortic smooth muscle cells. The authors concluded these approaches will help to achieve reproducible results and allow result comparison in a consistent way.

The data presented in this manuscript are interesting for terms of analyses of vascular calcification processes. However, the reviewer has some major remarks:

Major points:

1. The introduction section has to be revised. The authors should focus on the main topic of the manuscript and should avoid an very general description of the methods available for the investigation of vascular calcification processes (eg.g page 3 , line 56-59).

The introduction has been revised following the suggestion of the reviewer and two references have been added.

2. The authors should prepare a table demonstrating the condition analyze within the study.

A supplementary table including all the culture conditions as required by the reviewer has been added to the manuscript (See the S3_culture conditions in supplemental data).

3. NG, KM, FD and AA are obviously employees of a company (RD-Nephrologie). A statement has to be added whether these authors have to declare a conflict of interest based on the employment.

NG, KM FD and AA are employees of RD – Néphrologie and have declared to have no conflict of interest.

4. Figure 3.B: A bar graph demonstrating the quantification of Phosphosmad1 should be added.

The bar graph requested by the reviewer with the quantification of phosphosmad1 has been added to the figure 3B and the complete data have been included in the supplemental data file, S2_Table Data

5. Not significant data has to be removed from the revised version of the manuscript (page 12, line 258 - 260)

The paragraph alluded to by the reviewer has been removed as requested 

6. The authors should discuss the overall aim of the study in the discussion section in more detail.

The overall aim of the manuscript has been now better presented in the discussion section of the new version of the manuscript.

7. Some typos should be removed in the revised version of the manuscript, e.g. page 9, line 169 (3.8mM instead of 3.8 mM). 

The typos have been corrected

Reviewer #2: This experiment allows overcoming of an ethical issue for using animal models in experiments of vascular calcification research as one of the most important outcome for experiencing major cardiovascular events in humans as a topic for research, although the authors well pointed out the extrapolation of the data should be taken carefully into account.

Reviewer #3: The aim of this study is to analyze in vitro and ex vivo models of vascular calcification and to select factors that may affect the process of calcification.

Th study is well performed and I cannot identify methodological defects

However there are points that should be clarified

-Fig1 shows that calcium deposits in HASMCs cultured with high phosphate

(3.8 mM) are decreased by supplementing with 15% FCS. It is important to determine if ionized calcium and the concentration of phosphate are modified by 15% FCS. It is know that FCS contains large proteins “calciproteins” that may substract Ca and P from the solution. At least the authors should evaluate if for a given pH, FCS modifies the Ca x P product.

The remark of the reviewer is very pertinent. We checked, as suggested by the reviewer, the pH and the content of calcium and phosphate in both, serum free and serum supplemented media. The pH was 7.6 in both media showing that the DMEM medium used is highly buffered. Concerning the calcium content it was 1.9mM and 2.2mMy while the phosphate was 3mM and 3,25mM without and with 15% FCS respectively. The Ca x P product was 5.7 and 7.2 in 0% and 15% FCS medium respectively. Unfortunately, we were not able to measure the ionised calcium, as we do not have specific electrodes to do so. Therefore, the putative substraction of calcium and phosphate from the culture medium by the serum components added in 15% FCS conditions was not observed in our experiments. By the contrary, the slight increase in Ca x Phos product might be the result of the addition of the calcium and phosphate contained in the FCS.

-The total calcium content in the wells was significantly increased in the

wells with cells cultured with the basic P donor only (P < 0.001) vs acidic and neutral

conditions. It is likely that the high pH is increasing calcium deposit the opposed is expected to occur in acid pH. Again ionized Ca and pH should be measured minute after the addition of the different forms of phosphate.

The pH of culture medium was 7,6 for all the medium and was not modified by P donors. Unbound calcium was also measured and no difference was observed between the media with different P donors. The results are given in the S4_Medium calcium concentration of the open data.

-It seems that FCS is required by alkaline phosphatase to increase activity on calcium deposition and that this enhancing activity may be independent of the presence of cells (Fig 2D). It will be important to analyze if FCS contains pyrophosphate; hydrolysis of pyrophosphate by alkaline phosphatase may affect calcification.

The interaction between pyrophosphate and alkaline phosphatase is an interesting and probably very complex point. We included alkaline phosphatase effects on the tested media, both for cell culture systems and aortic explants on that basis. However, the surprising effects observed in both systems, with the decrease in the calcification of the aortic explants in the presence of alkaline phosphatase, led us to remove it from the proposed medium, as already stated in the discussion section (page 19). As underlined by reviewer #1 the main aim of the present paper was to find a calcifying system optimised to be proposed to a wider group of researchers interested in calcification processes. 

The interaction between pyrophosphate and alkaline phosphatase may be proposed as a separate topic to be explored in a completely different project. 

-The authors state the optimal culture conditions to obtain medial calcification of aortic rings were 15% FCS supplemented medium without alkaline phosphatase. Thus data should be contrasted with results by KI 2008 73(9):1024.

The article of Lomashvili, has been commented (reference #28) and the results contrasted with our findings in the new version of the manuscript. 

Reviewer #4: The authors address a key problem in calcification research, which is the lack of reproducibility of in vitro calcification set ups. I have only minor comments.

1. What was the cell culture media pH for different phosphate donors used to induce calcification?

The pH was 7.6 for all the culture media except for the culture media without FCS the pH was 7.9.

2. Please indicate the number of times that experiments were repeated and provide DS for graphs without DS

The complete data of all the experiments performed are now available in the supplemental data (S2 of the present version). In it it can be found the number of repeats for all the experiments. 

3. Specifically, for the optimal ACM, please indicate whether every time the experiment was repeated, calcification was obtained or whether there are still some uncontrolled factors that may prevent calcification in certain experiments.

Calcification was obtained every time when using optimal ACM. However, the amount of calcium and calcium deposits may vary when changing the batch of FCS. 

4. Please in the discussion, add the concentrations of calcium and phosphate used in mg/dl, which are commonly used units in many countries, so as readers can easily contextualize to the clinical situation. Since calcium binds albumin and albumin levels in 15% FCS are lower than in 100% human serum, please discuss what would be the free calcium and total calcium equivalent of the concentrations used in experiments.

We added the equivalent concentration in the discussion in mg/dL in brackets. 

We understand the willingness of the reviewer in trying to make an easy parallelism between the proposed experimental systems and the clinical situation. However, we have to keep in mind that the in vitro and ex vivo settings are far from the in vivo and clinical situations and trying to make any equivalence is a hazardous approach that may be misleading and we would like to avoid if possible.

---

## [Decision Letter · Decision Letter 1]

25 Feb 2020

Optimisation of cell and * ex vivo * culture conditions to study vascular calcification

PONE-D-19-33486R1

Dear Dr. gayrard,

We are pleased to inform you that your manuscript has been judged scientifically suitable for publication and will be formally accepted for publication once it complies with all outstanding technical requirements.

With kind regards,

Harald Mischak

Academic Editor

PLOS ONE

Additional Editor Comments (optional):

Reviewers' comments:

Reviewer's Responses to Questions

**Comments to the Author**

1. If the authors have adequately addressed your comments raised in a previous round of review and you feel that this manuscript is now acceptable for publication, you may indicate that here to bypass the “Comments to the Author” section, enter your conflict of interest statement in the “Confidential to Editor” section, and submit your "Accept" recommendation.

Reviewer #3: All comments have been addressed

Reviewer #4: All comments have been addressed

2. Is the manuscript technically sound, and do the data support the conclusions?

Reviewer #3: Yes

Reviewer #4: Yes

3. Has the statistical analysis been performed appropriately and rigorously? 

Reviewer #3: Yes

Reviewer #4: Yes

4. Have the authors made all data underlying the findings in their manuscript fully available?

Reviewer #3: Yes

Reviewer #4: Yes

5. Is the manuscript presented in an intelligible fashion and written in standard English?

Reviewer #3: Yes

Reviewer #4: Yes

6. Review Comments to the Author

Reviewer #3: All comments have been addressed. I do not think that additional experiments are required. The data definitely support the conclusion

Reviewer #4: (No Response)

7. PLOS authors have the option to publish the peer review history of their article (what does this mean?). If published, this will include your full peer review and any attached files.

Reviewer #3: No

Reviewer #4: No

---

## [Editor Report · Acceptance letter]

27 Feb 2020

PONE-D-19-33486R1 

Optimisation of cell and * ex vivo * culture conditions to study vascular calcification 

Dear Dr. gayrard:

I am pleased to inform you that your manuscript has been deemed suitable for publication in PLOS ONE. Congratulations! Your manuscript is now with our production department. 

With kind regards,

on behalf of

Prof. Harald Mischak 

Academic Editor

PLOS ONE